# Reconfigurable MEMS Fano metasurfaces with multiple-input–output states for logic operations at terahertz frequencies

Manukumara Manjappa [1,2], Prakash Pitchappa[1,2], Navab Singh[3], Nan Wang[3], Nikolay I. Zheludev[2,4], Chengkuo Lee [5,6] & Ranjan Singh [1,2]

A broad range of dynamic metasurfaces has been developed for manipulating the intensity, phase and wavefront of electromagnetic radiation from microwaves to optical frequencies. However, most of these metasurfaces operate in single-input–output state. Here, we experimentally demonstrate a reconfigurable MEMS Fano resonant metasurface possessing multiple-input–output (MIO) states that performs logic operations with two independently controlled electrical inputs and an optical readout at terahertz frequencies. The far-field behaviour of Fano resonance exhibits XOR and XNOR operations, while the near-field resonant confinement enables the NAND operation. The MIO configuration resembling hysteresis-type closed-loop behaviour is realized through inducing electromechanically tuneable out-of-plane anisotropy in the near-field coupling of constituent resonator structures. The XOR metamaterial gate possesses potential applications in cryptographically secured terahertz wireless communication networks. Furthermore, the MIO features could lay the foundation for the realization of programmable and randomly accessible metamaterials with enhanced electro-optical performance across terahertz, infrared and optical frequencies.

[1] Division of Physics and Applied Physics, School of Physical and Mathematical Sciences, Nanyang Technological University, 21 Nanyang Link, Singapore 637371, Singapore. [2] Centre for Disruptive Photonic Technologies, The Photonics Institute, Nanyang Technological University, 50 Nanyang Avenue, Singapore 639798, Singapore. [3] Institute of Microelectronics, 11 Science Park Road, Singapore 117685, Singapore. [4] Optoelectronics Research Centre and Centre for Photonic Metamaterials, University of Southampton, Highfield, Southampton SO17 1BJ, UK. [5] Department of Electrical & Computer Engineering, National University of Singapore, 4 Engineering Drive 3, Singapore 117576, Singapore. [6] Center for Intelligent Sensors and MEMS (CISM), National University of Singapore, E6 #05-11F, 5 Engineering Drive 1, Singapore 117608, Singapore. Correspondence and requests for materials should be addressed to R.S. (email: ranjans@ntu.edu.sg)

Recent trends in the metamaterial research have advanced towards the realization of functional and reconfigurable metasurfaces[1–3] that enable real-time control over their geometrical and optical properties, thereby creating exceptional opportunities in the field of active and tuneable metamaterials. Over the years, various approaches have emerged in realizing tuneable metamaterials through reconfiguring their structure and geometry via external stimulus such as electrical control[4–7], magnetic field[8,9], thermal gradient[10–12] and optical pulse[13–18]. A specific class of structurally reconfigurable metasurfaces based on micro/nano electromechanical systems (MEMS/NEMS) have given a unique advantage for active manipulation of the near-fields in all the three spatial directions by exploiting sensitive changes to their micro/nano scale movements. Near-fields[19] are the most significant components of the scattered fields that stay closer to the object surface and fail to radiate freely to the far-field. The omnipresent nature of the near-field and the finest information that it entraps, makes it a vital component in the light–matter interactions. Therefore, dynamic control over the near-fields provides a new paradigm of manipulating the light–matter interactions, which makes them more resilient and merits their applications in future generation state-of-the-art active photonic devices. At the terahertz (THz) and infrared frequencies, the MEMS/NEMS metasurfaces have enabled dynamic manipulation of near-field entities thereby showing an active reconfiguration of intriguing features like magnetic response[4,10], transparency[20], near-perfect absorption[21], phase engineering[22], resonance modulation[23], anisotropy[5] and THz invisibility[24]. However, apart from these useful advancements, the ability to control and tailor the near-field interactions by establishing multiple controls at the unit-cell level has remained elusive.

The bright prospects of functional metamaterials lie in achieving multiple controls within the unit cell of the metamaterial, which could provide a flexible platform for realizing extremely versatile devices manifesting enhanced electro-optical performance. Multiple controls within the unit cell would enable precise tailoring of near-field interactions between the meta-atoms by relatively manoeuvring their structural properties, thereby obtaining the optical properties on demand[25]. Recently, the MEMS/NEMS switchable metamaterials[2,7] have provided a promising pathway to control near-field coupling between the metamolecules by establishing independent/multiple controls over the structural reconfiguration of the constituent resonators within the unit cell of the metamaterials. However, most of these metamaterial designs operate in single output configurations. In the hindsight of enhancing their multifunctional capabilities in digital[26–28] and multichannel signal processing applications, one way is to establish a multi-valued dependency between the input and output characteristics of the metamaterial. In the past, the multiple output states signifying the hysteresis behaviour has been shown in various hybrid metamaterial systems composed of vanadium dioxide ($VO_2$)[12] and graphene–ferroelectric polymer[29] that demonstrated the memory effects and logic gate functionalities at THz frequencies. However, these results were based on single input control and depend on the properties of the integrated natural material that dictates their output efficiencies along with limiting their operation to specific frequencies.

Here, we experimentally realize the excitation and active tuning of sharp Fano resonances in a MEMS reconfigurable metasurface exhibiting multiple-input–output (MIO) characteristics in its near-field as well as far-field optical properties. These MIO states are created by establishing anisotropic nature in the near-field coupling between the asymmetric resonators in the out-of-plane (z-axis) reconfigurable metasurface that excites a sharp Fano-type resonance feature. The reconfigurable geometry of the MEMS Fano-metasurface provides various structural meta-stable states by using two independently controllable electrical inputs and an optical/near-field readout that enables the realization of digital logic gates. Here, we demonstrate exclusive-OR (XOR), XNOR and NOT logic gates in the far-field optical states, and the NAND logic gate operation in the near-field characteristics of the device at THz frequencies. Fundamentally, XOR is an important secondary Boolean (logic) operation that is a composite of the basic logic functions and is not linearly separable. This property makes it more resilient and practically useful in the information and computational technologies as parity generators, one-time pad (OTP)-based unbreakable cryptography protocols[30,31], pseudorandom number generators and digital encoders or decoders in signal processing. On the other hand, the NAND logic operation is a functionally complete set logic operator which can be used to express all the basic set logic operations by defining a network of NAND gates. The multiple-logic operations together with the volatile and nonvolatile[32,33] regimes of MEMS actuation can enhance the digital functionalities of the device in realizing optical memory registers to encode, harvest, process and send secured information in the form of encoded/decoded optical bits at THz frequencies.

## Results

**Design and fabrication**. To precisely elucidate the operation of logic gate functionalities through active control of Fano resonances, we fabricated a MEMS-based metasurface consisting of two split ring resonators (we term them SRR-1 and SRR-2) that are independently and sequentially actuated by applying the voltages $V_1$ and $V_2$ (shown in Fig. 1a). The device is fabricated using the photolithography technique, where periodic array of bimorph SRRs (900 nm thick aluminium (Al) deposited on top of 50 nm aluminium oxide ($Al_2O_3$) layer) possessing mirror-symmetry are patterned on a lightly doped silicon (Si) substrate (refer to the Methods section and Supplementary Figs. 1–5 for the device fabrication details and characterization). Due to the residual stress in the bimorph layers, the cantilevers are bent up, thereby increasing their released heights ($h$). Scanning electron microscope (SEM) image of the fabricated MEMS Fano-metasurface is shown in Fig. 1a in the coloured scale that illustrates the maximum asymmetric state of the device with SRR-1 snapped down on the substrate using voltage $V_1 = 35$ V and SRR-2 is retained in the released state of the bimorph cantilevers with $V_2 = 0$ V. The out-of-plane reconfiguration of the released cantilevers is achieved through electrostatic actuation, by applying voltage across the Al layer and silicon substrate. Metal lines connecting the SRR-1 and SRR-2 cantilevers are electrically isolated from each other and this allows for the independent reconfiguration of heights $h_1$ and $h_2$ through the application of voltages $V_1$ and $V_2$, respectively. The selective reconfiguration at the sub-unit cell level provides the flexibility to introduce dynamically tuneable structural asymmetry along the out-of-plane (z)-axis of the sample. The out-of-plane structural asymmetry parameter is defined as $\delta = \frac{|h_1 - h_2|}{s} \times 100\%$, where $s$ is the length and $h_1$, $h_2$ are respectively the released heights of SRR-1 and SRR-2 cantilever arms. Figure 1b–d are the SEM images of the unit cell showing the sequential control of SRR cantilevers by applying voltages $V_1$ and $V_2$ across the Al-metal lines and the silicon substrate.

**Active control of Fano resonance**. Persistent control of resonance features in the MEMS Fano-metasurface is experimentally characterized by using the photoconductive antenna-based THz-time domain spectroscopy setup in the transmission mode (refer to the Methods section for more details on experimental

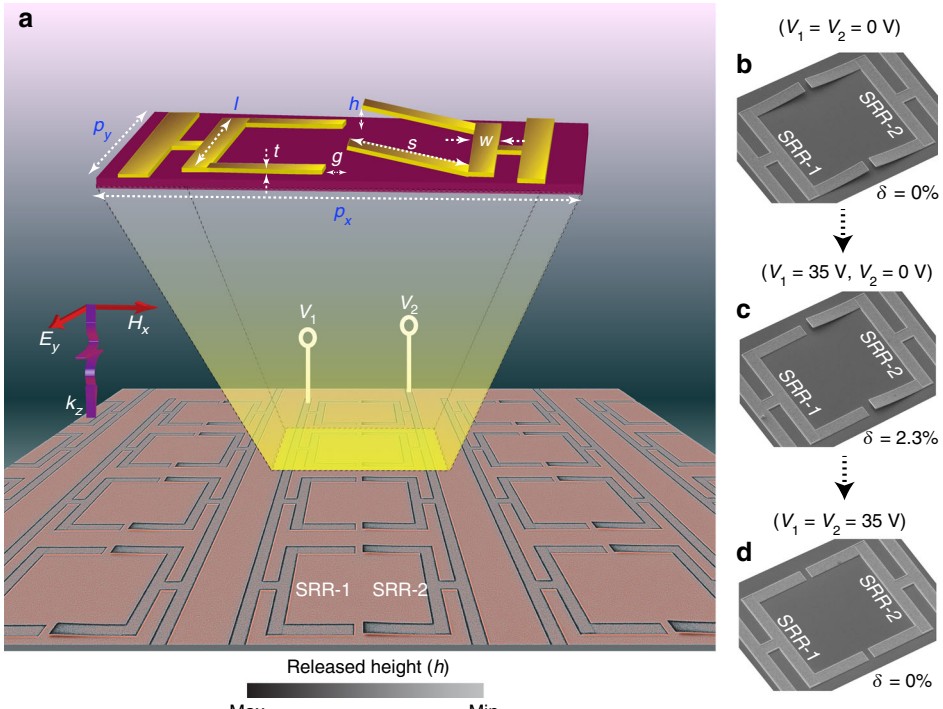

**Fig. 1** Fabricated sample images and the sequential operation of the device. **a** Coloured scanning electron microscope (SEM) image of the MEMS Fano-metasurface. The unit cell comprises of two SRRs separated by a gap $g$, where SRR cantilever arms of length $s$ are released at a height $h$. The unit cell dimensions are depicted in the inset, where $p_x$: 110 μm; $p_y$: 75 μm; $l$: 60 μm; $s$: 25 μm; $w$: 6 μm; $g$: 4 μm; and $t$: 900 nm. $V_1$ and $V_2$ are the input voltage ports to achieve the independent actuation of SRR-1 and SRR-2, respectively. **b**–**d** SEM images of the unit cell showing the sequential actuation of SRRs with voltage $V_1$ and $V_2$ applied across the two SRRs, where the sequence from (**b**) to (**c**) represents the increasing asymmetry ($\delta$) and (**c**) to (**d**) represents the decreasing asymmetry configuration

procedure). The measured transmission spectra for increasing voltage $V_1$ with $V_2 = 0$ V are shown in Fig. 2a. The inset diagram in Fig. 2b presents the experimentally measured mechanical deformation profile of the cantilever by applying the voltage on one of the SRRs (details on the device characterization are given in the Methods section). Initially, for the case where no voltage is applied across the resonators, i.e. $V_{1,2} = 0$ (see Fig. 1b), the cantilever arms of the two SRRs are symmetrically inclined at same heights $h_1 = h_2 = h$ ($\delta = 0$) along the $z$-axis. Such symmetric configuration of resonators results in the excitation of strong dipole type of resonance at 0.77 THz for the incident THz radiation polarized in the $\mathbf{E_y}$ direction. When voltage ($V_1$) is applied across the Al lines of the released cantilevers (say, SRR-1) and Si substrate, the suspended SRR-1 cantilevers gradually bend towards the substrate due to the attractive electrostatic force. This deformation in the height of the SRR-1 cantilevers creates a structural asymmetry ($\delta$) along the $z$-axis of the metasurface sample. As a result, near-field coupling between the asymmetric structures excites a sharp and weak Fano resonance[34–37] feature (at 0.58 THz, red curve in Fig. 2a) within a broad dipolar resonance. Upon continuously increasing $V_1$ across the SRR-1, strength of Fano resonance grows and reaches its maximum amplitude for $V_1 = 35$ V (at 0.56 THz, where $V_2 = 0$ V), with a slight red shift in its resonance frequency. Subsequently, when the voltage $V_2$ is applied across SRR-2 by keeping SRR-1 on the substrate, the cantilever arms of SRR-2 are gradually pulled towards the substrate, which decreases the asymmetry in the system. Due to this decrease in asymmetry, the Fano resonance starts to weaken with increasing $V_2$ and completely diminishes at $V_2 = 35$ V, as shown in Fig. 2b (both SRR-1 and SRR-2 are snapped down on the substrate with $V_{1,2} = 35$ V (see Fig. 1d)). Thereby, the symmetry of the structures is restored in the system

that now shows only a dipolar resonance (at 0.67 THz). The observed red-shift in the frequency of the dipole and the Fano resonance is due to enhanced capacitance in the air gap between the cantilever and the substrate, as the cantilever is gradually bent down onto the substrate. We also fabricated the samples with various metal thicknesses of cantilevers (300, 500 and 700 nm) possessing different released heights. The THz transmission measurements were performed on the samples and are shown in Supplementary Fig. 4. The cantilever with thinner metal film possesses larger release height ($h$) (see Supplementary Fig. 5), and hence aids in achieving larger structural asymmetry ($\delta$) in the system, which in turn results in stronger Fano resonance amplitude. The main reason for using the 900 nm thick aluminium resonator sample for our detailed analysis is the enhanced structural stability offered by thicker cantilevers during the persistent tuning of their released heights. This factor aids in precise control and continuous active tuning of Fano resonance feature in the proposed MEMS Fano-metasurface structure.

The correspondence between the transmission spectra obtained by the sequentially applied voltages ($V_1$ and $V_2$) and the structural asymmetry ($\delta$) of the MEMS Fano-metasurface is established by the numerical simulations. The transmission spectra for varying $\delta$ are shown in Fig. 2c, d, which are calculated using finite difference time domain (FDTD) simulations offered by the commercially available Computer Software Technology (CST) microwave studio using unit-cell boundary conditions (refer to the Methods section for more details). The value of structural asymmetry parameter ($\delta$) is estimated using the defined expression for $\delta$ based on the experimentally measured inclined heights ($h_1$, $h_2$) of the cantilevers of resonators SRR-1 and SRR-2 (as shown in the inset of Fig. 2b). The insets in Fig. 2c, d represent the sequential actuation of SRR-1 and SRR-2 resonators that

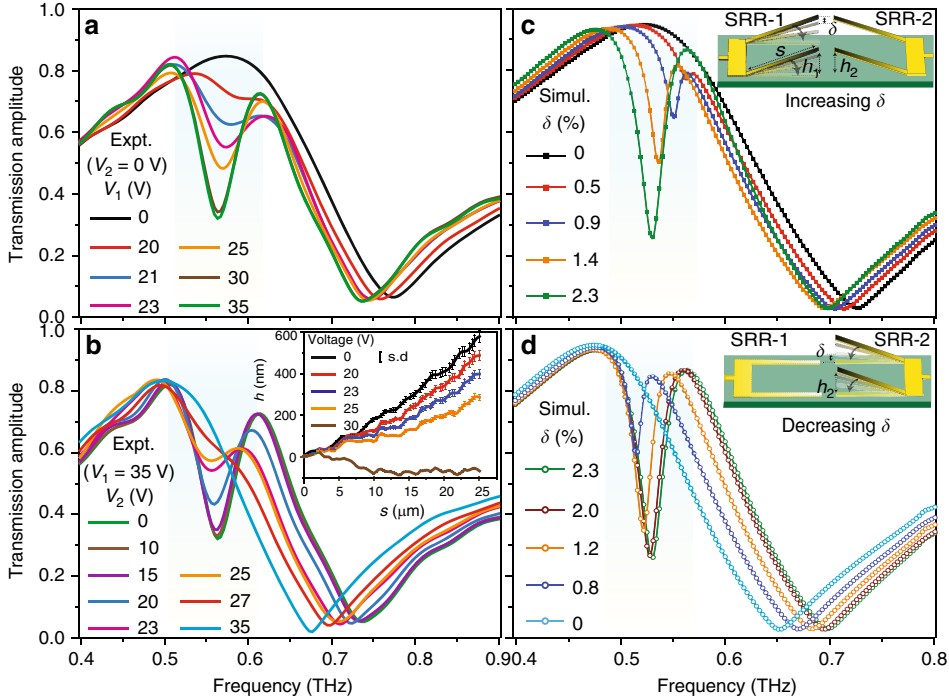

**Fig. 2** Active tuning of Fano resonances in MEMS metasurface. **a** Depicts the experimentally measured THz transmission spectra showing the evolution of Fano resonance for continuous actuation of SRR-1 by varying voltage $V_1$, while keeping $V_2 = 0$ V. **b** Represents the measured spectra resulting from the actuation of SRR-2 by increasing $V_2$, while keeping $V_1 = 35$ V. Inset figure depicts the experimentally mapped actuation angles (with the error bars) of the SRR cantilevers of metal thickness 900 nm under the applied voltage ($V$) for the designed MEMS Fano-metasurface. **c**, **d** Numerically simulated THz transmission spectra for increasing and decreasing structural asymmetry configurations of the proposed MEMS Fano-metasurface. The values of the depicted asymmetry parameter ($\delta$) in (**c**) and (**d**) show one-to-one correspondence with the voltage values of $V_1$ and $V_2$ varied in (**a**) and (**b**), respectively. The insets in (**c**) and (**d**) signify the sequential actuation of SRR-1 and SRR-2, respectively, showing the increasing and decreasing structural asymmetry configurations

correspond to the continuous increase and decrease in $\delta$, which signifies one complete ramp cycle of asymmetry parameter ($\delta$). As a first actuation sequence, asymmetry in the structure is increased by decreasing the released height ($h_1$) of SRR-1, which results in strengthening of the Fano resonance feature that reaches its largest resonance amplitude at the maximum asymmetry of $\delta_{\max} = 2.3\%$. In the next actuation sequence, upon decreasing the released height ($h_2$) of SRR-2, amplitude of the Fano resonance diminishes and eventually disappears as the cantilever of SRR-2 touches down on the substrate ($\delta = 0$). Therefore, sequential actuation of the resonators SRR-1 and SRR-2 alters the symmetry of the structure from a symmetric configuration to an intermediate asymmetric state and finally bringing it back to the symmetric state thereby controlling the excitation of Fano resonance in one complete ramp cycle of $\delta$.

**Multiple-input–output (MIO) characteristics**. The most striking feature of the excitation of the Fano resonance in MEMS Fano-metasurface is the observed anisotropic nature in the near-field coupling between the resonators at a given asymmetry parameter ($\delta$). The anisotropic Fano coupling exhibits two distinctive pathways (in the output states) for the far-field and the near-field optical characteristics with respect to the sequential application of two voltage inputs $V_1$ and $V_2$, respectively on SRR-1 and SRR-2 or for increasing and decreasing pathways of $\delta$. The distinctive pathways in the far-field response of the device are analysed using the peak to peak transmission intensity ($|\Delta T|$) of Fano resonance, which is discussed in Supplementary Fig. 6. As a input control parameter, we define the sequential voltage operation of $V_1$ and $V_2$ in terms of the differential voltage applied between the two

resonators SRR-1 and SRR-2, defined as $\Delta V = |V_1 - V_2|$ that directly corresponds to the asymmetry parameter ($\delta$) of the structure. We plot the variation in $|\Delta T|$ with respect to increasing and decreasing values of $\Delta V$ applied across the resonators in Fig. 3a, where the $|\Delta T|$ exhibits distinctive values for the increasing and decreasing configuration of input parameter $\Delta V$. This scenario showing the distinctive pathways for $|\Delta T|$ closely resembles the hysteresis-type behaviour as observed in many natural phase change materials such as $VO_2$[12] and Ferrites[29]. However, here it is indeed two stable output states observed for two input controls ($V_1$ and $V_2$) that forms a closed loop in $|\Delta T|$ with respect to $|\Delta V|$ signifying the multiple-input–output states in the electro-optical characteristics of the metasurface. This hysteresis-type behaviour constituting MIO states is artificially created by the induced anisotropic near-field coupling observed in the tuning of Fano excitation by varying $\delta$ or $\Delta V$. Further, similar MIO states are observed for the measured $Q$-factors of Fano resonance features for increasing and decreasing values of $\Delta V$, as shown in Fig. 3c. We observe that for a given value of $\Delta V$, the $Q$-factors during the increasing configuration of $\Delta V$ follow a different variation and possess larger values compared to the decreasing pathway of $\Delta V$. The maximum $Q$-factors of 19.73 and 19 are experimentally measured for the lower $\Delta V$ (i.e. for extremely small asymmetry parameter ($\delta$) cases), respectively during the increasing and decreasing configuration of $\Delta V$. In the numerical simulations, the differential voltage ($\Delta V$) is expressed in terms of asymmetry ($\delta$) of the structure, which is a critical parameter in controlling the nature and excitation of Fano resonances. In Fig. 3b, d, the peak to peak transmission intensity ($|\Delta T|$) and $Q$-factors of Fano resonance are plotted for the

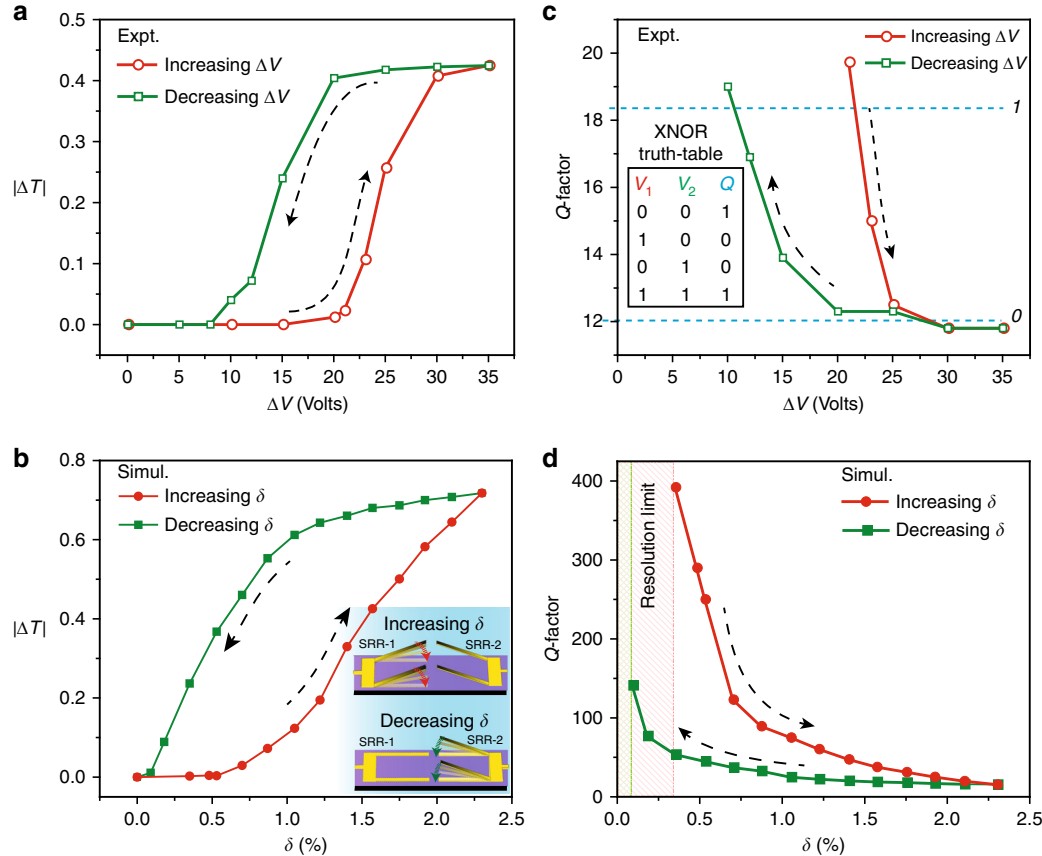

**Fig. 3** Multiple-input–output electro-optical characteristics in the far-field features of MEMS Fano-metasurface. **a** Measured Fano resonance transmission intensity ($|\Delta T|$) with respect to differential voltage ($\Delta V$) calculated for the curves shown in Fig. 2a and b, respectively. The red circles represent increasing order of Fano resonance strength by increasing the $\Delta V$ applied on the structures (applying $V_1$ on SRR-1 with $V_2 = 0$ V), whereas green squares represent decreasing state of Fano resonance in the presence of $V_2$ on SRR-2 with $V_1 = 35$ V that decreases the $\Delta V$ applied on the structure. **b** Simulated Fano resonance intensity ($|\Delta T|$) showing two intensity states for a single asymmetry value ($\delta$) of the system. The inset figure represents the sequential actuation of SRR-1 and SRR-2 that governs the observed multiple-input–output (MIO) states for the MEMS Fano-metasurface. **c** Experimentally and **d** numerically calculated $Q$-factors of the Fano resonance are shown that exhibits the MIO configuration for sequential actuation of the SRR-1 and SRR-2 resonators. Inset table in (**c**) represents the truth-table for the logic exclusive-NOR (XNOR) operation that can be visualized in the proposed MEMS Fano-metasurface device in the form of high ('1') and low ('0') values for $Q$-factors of Fano resonance in the far-field

increasing and decreasing scenario of structural asymmetry parameter ($\delta$), where the SRR-1 and SRR-2 are actuated sequentially (as shown in the insets of Fig. 3b). The simulation results agree well with the MIO characteristics observed in the experimental data. Further detailed plots on the figure of merit[38] (FoM) of Fano resonance showing the MIO characteristics are provided in Supplementary Figs. 10 and 11. These MIO states enacted by the out-of-plane (three-dimensional) symmetry breaking induced Fano resonances provide a unique advantage of creating a closed-loop behaviour in the electro-optical properties with tuneable area under the loop, which could help in precise tailoring of energy dissipation in the system. The tuneable MIO states for the intensity, $Q$-factors and FoM of Fano resonance are depicted in Supplementary Figs. 12–14. This tuneable MIO feature could lead towards the realization of multiple (more than two) output states favouring the possibility of digitizing the optical response through the system with two (or more) independent input-voltage control parameters.

**MEMS Fano-metasurface enacted logic gates**. The uniqueness of digitizing the excitation of Fano resonance in terms of its far-field optical states using two electrical controls constitutes digital XOR, XNOR, PASS and NOT logic functions in the far-field spectrum

at THz frequencies. As discussed earlier, the structural states ('up' or 'down') of the constituent resonators SRR-1 and SRR-2 are independently reconfigured using the voltage inputs $V_1$ and $V_2$, respectively in determining the output state of Fano resonance ($F$). The structural metastable states of the resonators determined by the electrical inputs ($V_{1,2}$) are represented by the logic binary digits, where 'up-state' of the resonator corresponds to binary '0' ($V_{1,2} = 0$ V) and the 'down-state' corresponds to binary '1' ($V_{1,2} = 35$ V). True (ON) and false (OFF) states of the Fano resonance amplitude in the far-field are represented by the binary digits '1' and '0', respectively. The measured THz far-field transmission spectra showing the XOR logic feature for the various metastable structural states (00), (10), (01) and (11) of the MEMS resonators are given in Fig. 4a–d. The voltage inputs ($V_1$ and $V_2$) applied to the individual resonators (SRR-1 and SRR-2) can be programmed using sequential trigger bits {0,1} that controls the actuation heights (up/down) of SRR-1 and SRR-2, respectively. However, in our experiments, the limitations posed by the well-known problem of stiction[39] in our fabricated MEMS devices disrupt the repeatable operation of the device. Hence, we consider two devices and prepare their cantilevers in their respective metastable states (10) and (01) by applying the corresponding voltage inputs and perform the transmission measurements in the far-field. Since the Fano resonance feature

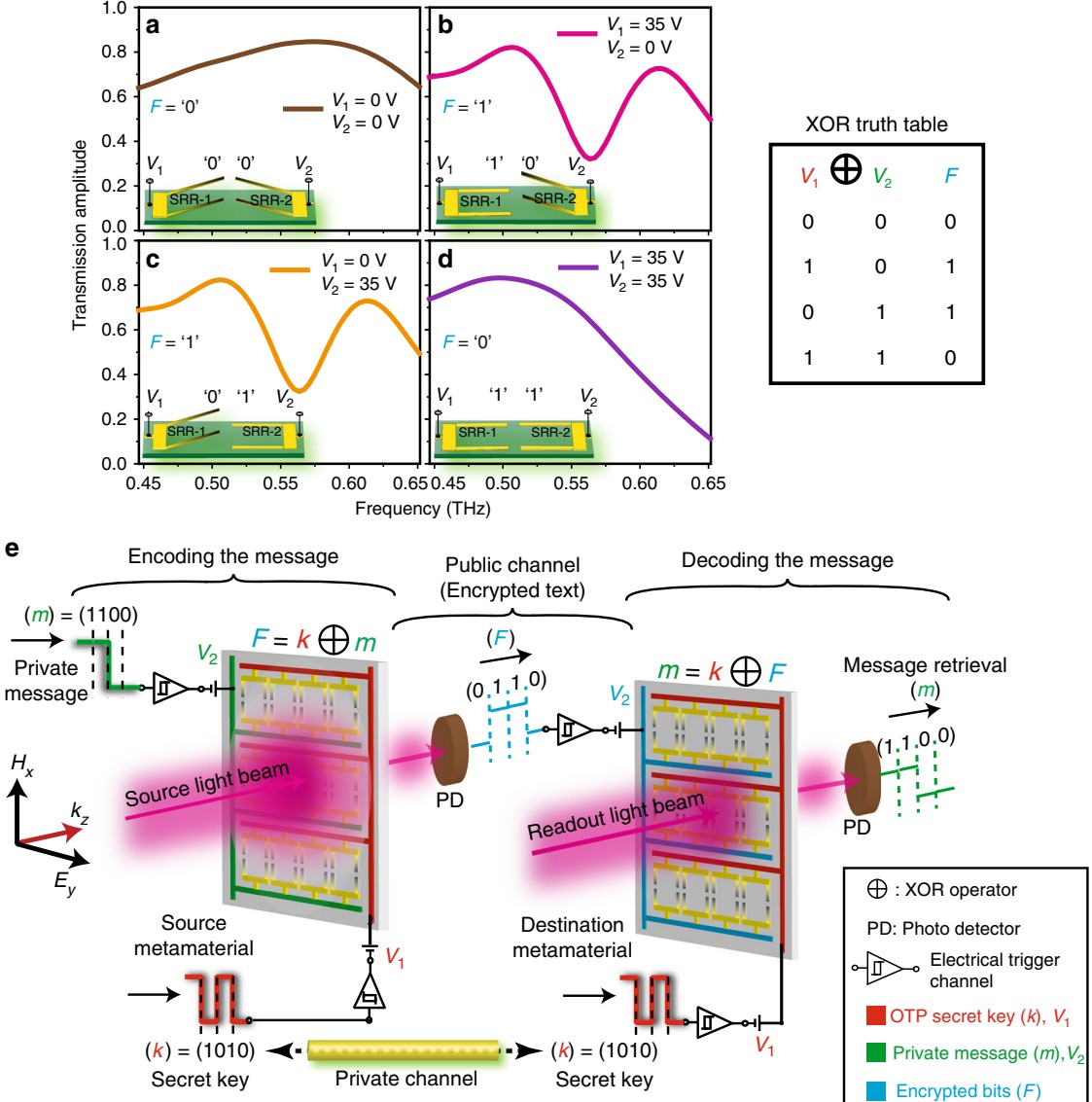

**Fig. 4** Exclusive-OR (XOR) logic operation with MEMS Fano-metasurface and its significance in cryptographic wireless communication networks. **a–d** Measured far-field THz transmission spectra of the MEMS Fano-metasurface showing the XOR logic feature in the form of presence/absence of Fano resonance (at 0.56 THz) for various structural/voltage states of the SRRs. **a, d** show the symmetric configuration of the structures (*00* and *11*) that signifies the absence of Fano resonance excitation ($F = 0$), whereas **b, c** represent the asymmetric configuration of the structures (*10* and *01*) that results in the excitation of Fano resonance feature ($F = 1$) in the sample. **e** Pictorial representation of realizing the OTP secured wireless communication channel by performing the XOR logic operation to encode the private message (*m*) with the secret key (*k*) and is sent through the public channel as optical signals. The message is retrieved securely (decrypted) at the destination end by performing the inverse XOR operation on the measured optical states (*F*) with the secret key (*k*). The structural/voltage states of SRR-1 and SRR-2 are expressed as OTP secret key (*k*) and the private message (*m*), respectively, whereas the secured data is transmitted through public channel in the form of optical bits (*F*)

results due to the asymmetry in the structural configuration of the metasurface, for input voltages ($V_1 = 0$, $V_2 = 35$ V and $V_1 = 35$ V, $V_2 = 0$ V) there exist two asymmetric structural configurations 'up–down' (*01*) and 'down–up' (*10*) that results in the true state for the Fano resonance condition (i.e. $F = 1$), as shown in Fig. 4b, c, respectively. On the other hand, for symmetric structural configurations of the device ($V_1 = V_2 = 0$ V and $V_1 = V_2 = 35$ V), 'up–up' (*00*) and 'down–down' (*11*) results in the absence of Fano resonance ($F = 0$) state, as shown in Fig. 4a, d. Resulting truth table is presented in the inset of Fig. 4, which resembles the digital XOR logic operation, where the Fano output is true ($F = 1$) if the input voltage states differ, otherwise Fano output results in a false state ($F = 0$) (i.e. when both the inputs are either true (*11*) or false (*00*), then output state $F = 0$). As discussed in Supplementary

Fig. 7, intensity contrast between the output states $F = 0$ and $F = 1$ is measured to be equal to $\Delta T = 0.61$ (normalized to the input intensity of 1) at a frequency of 0.56 THz. The proposed XOR logic functionality based on the Fano resonance exhibits substantial improvement in the intensity contrast ratio compared to earlier demonstrations at THz frequencies[29,40]. Furthermore, another logic functionality that represents the XNOR truth table is derived from the MIO characteristics observed in the Q-factors of the Fano resonance, as shown in Fig. 3c. The XNOR logic operation signifies the true output if both the inputs are either true or false, which is complementary to the XOR logic operation. In our measurements, by defining the threshold values to the input voltages ($V_1$, $V_2$) and the output states in the form of Q-factors (inverse of loss-factors), we construct the XNOR

functionality of the device. Here, with either very low (*00*) or high (*11*) voltage inputs, the system exhibits extremely high values of *Q*-factors showing the true output state ($Q = 1$) for the low structural asymmetry region. In the case of large structural asymmetries, where one of the voltage input is high and other input is low (*10* or *01*), results in the false output state ($Q = 0$) with low *Q*-factors for the Fano resonance feature.

The proposed design further provides a flexibility of controlling the asymmetry of the structure to show either anisotropic or isotropic way of tuning the coupling between the adjacent resonators just by adequately coding the input electrical (voltage) signalling sequence. Supplementary Fig. 15 represents the experimentally measured variation in the intensity of the Fano resonance with respect to the voltage $V_2$ applied on SRR-2, by keeping SRR-1 in contact with the substrate with $V_1 = 35$ V (decreasing pathway of the asymmetry). This configuration of Fano resonance tuning signifies the single-input–output (SIO) characteristics in the MEMS Fano-metasurface, which is due to the isotropic nature of coupling between the resonators during the increasing and decreasing pathways of asymmetry ($\delta$). The ability of converting the electro-optical response of the system from MIO configuration (anisotropic tuning of coupling) to the SIO configuration (isotropic change in coupling) aids in realizing the NOT and PASS logic operations. By closing the input with high (*1*) and low (*0*) logic states, respectively and varying the other input, the pre-selection of the states with an added control enables a switch between NOT to PASS logic functions. These NOT and PASS logic operations are the special cases of XOR functionality of the MIO states. The NOT or negation operation represents the formation/annihilation of a Fano mode in the far-field spectrum of the device in the absence/presence of external stimulus, signifying the switching between the coupled and uncoupled regime of the metasurface system. On the other hand, by closing the input having the low ('*0*') logic state, the buffer/

PASS gate can be realized, where the operation does not alter the input state and hence the output logic state stays the same as the input logic state. The details on the NOT and PASS logic functionalities of the proposed design is provided in Supplementary Fig. 15.

In addition to the XOR, XNOR, NOT and PASS logic operations using the far-field characteristics, the near-field characteristics reveal the NAND logic operation in the form of confined electric fields in their ON (snapped) and OFF (released) states. Numerically calculated electric field amplitude distributions for various structural states of the MEMS Fano-metasurface are plotted in Fig. 5a–d. The absolute **E**-field amplitude for structural configurations shown in Fig. 5a–c represents enhanced field strengths when at least one of the structural states is in released (OFF) state compared to structural configuration (shown in Fig. 5d), where both cantilevers are prepared in snapped (ON) states. Variation in the amplitude of the confined electric fields is plotted in Fig. 5e that highlights the distinctive pathways for the increasing and decreasing configuration of the asymmetry. For the symmetric state with both the cantilevers in the OFF state (Fig. 5a), the electric field confinement is nearly an order of magnitude greater than the symmetric configuration, where both the cantilevers are in the ON states (Fig. 5d). Whereas, for the two asymmetric configurations of the cantilevers (Fig. 5b, c), the field confinement in the structure shows similar amplitude to the symmetrically prepared OFF states (Fig. 5a) of the structure. Higher amplitude value of the electric field confinement is labelled as binary '*1*', whereas lower electric field amplitude is represented by binary '*0*', as shown in Fig. 5e. Therefore, the change in confined near-field electric amplitude measured at the tip of the cantilevers in their various metastable structural states constitutes a logic NAND function as described by the truth table given in Fig. 5f. Experimental extraction of the near-field information in the THz part of the spectrum is challenging,

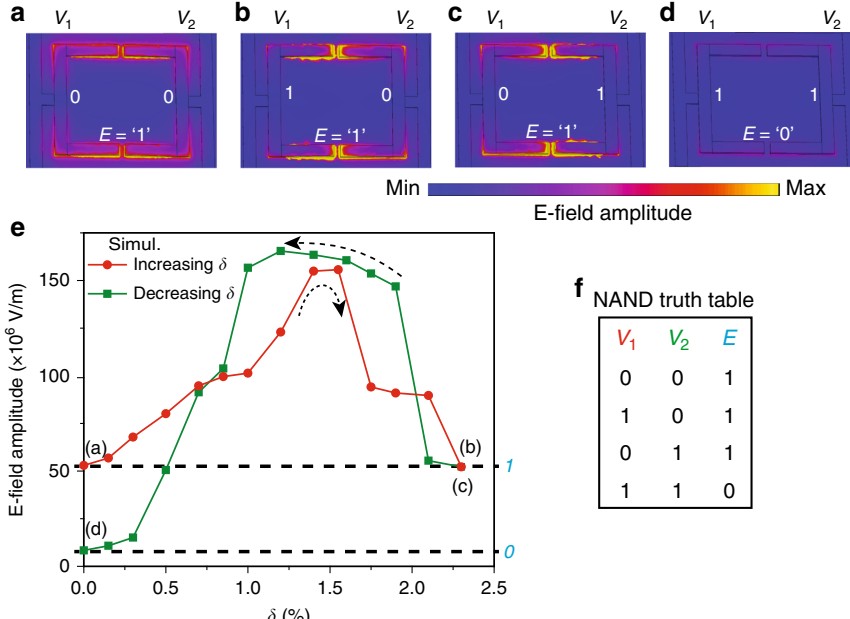

**Fig. 5** NAND logic gate and multiple-input–output states in the near-field confined energy of MEMS Fano-metasurface. **a–d** The numerically calculated electric field distributions at 0.55 THz, where **a–c** represent the true states of the electric field confinement ($E = 1$), whereas **d** shows the false state of $E$ (i.e. $E = 0$), signifying the construction of NAND logic operation that is tabulated in (**f**). The input logic states *0* and *1* for $V_1$ and $V_2$ represent the 'up' and 'down' actuation states of SRR cantilevers, respectively. **e** Distinctive variations shown for the enhanced spatially confined electric near-field strengths in the device at Fano frequencies (0.55 THz) that possess MIO behaviour during sequential actuation of the SRR-1 (increasing $\delta$) and SRR-2 (decreasing $\delta$). The electric field strength is measured at the tip position of the SRR cantilevers. **f** NAND logic truth table generated from the electro-optical operation of proposed MEMS Fano-metasurface

however, recent demonstrations using tip based THz near-field microscopy[41–43] can be used as one of the ways to retrieve the near-field information in terms of output field intensity states that in turn can be used to sequentially trigger the input voltage of the cascaded NAND device. This could enable the near-field cascading of the logic gates in the complementary metal-oxide-semiconductor (CMOS) configuration. Furthermore using numerical simulations, we conceptually show the NAND and OR logic operations in the far-field spectrum using the same geometrical design of MEMS Fano-metasurface but combining two unit-cells to make one composite unit-cell (super-cell) consisting of two Fano meta-molecules (a pair of SRR-1–SRR-2 and SRR-3–SRR-4), as shown in Supplementary Figs. 16 and 17. By independently controlling the SRRs present within the two Fano meta-molecules using the voltage sources ($V_1$ and $V_2$) the NAND and OR logic operations can be realized by the THz readout pulse in the form of presence or absence of Fano resonance ($F = 1$ or $0$) in the far-field amplitude or intensity spectrum. The NAND logic operation is significant owing to its unique feature of functional completeness (universal gate), as any Boolean function can be implemented by using the combination of several NAND gates, where the concept is schematically presented in Supplementary Fig. 18 to show the construction of AND and OR logic gates by cascading the NAND metasurface gates in the far-field.

## Discussion

The excitation of sharp Fano resonance feature in metamaterials has been pivotal in enhancing the confinement of near-field energy in the structures to aid the strong nonlinearity and sensing applications[36]. However, so far, such resonances have been realized by breaking the in-plane symmetry of the structures that was restricted to the SIO behaviour in the optical properties of the metamaterial. In this work, by breaking the structural symmetry in the out-of-plane (third) dimension of metamaterial allows us to probe the intriguing features resulting from the exponentially decaying nature of out-of-plane fringing near-fields. This non-linear spatial extension of near-fields in the third dimension of the sample aids in the anisotropic tuning of Fano resonance during its increasing and decreasing asymmetry ($\delta$) pathways, which leads to the observed MIO states in the excitation of Fano resonance (Fig. 3). This anisotropic type of Fano tuning is a result of contrasting coupling strengths between the resonators (SRR-1 and SRR-2) for same $\delta$, depending on whether the interacting resonator cantilevers are closer to or farther away from the surface of the substrate. For instance, in the region of low structural asymmetry ($\delta$), during the increasing $\delta$ configuration, the cantilevers of deforming SRR-1 and fixed SRR-2 are positioned away from the substrate, where the spatial extention of the near-field is weak (refer to inset of Fig. 3b). Therefore, the near-field interaction between the cantilever arms of SRR-1 and SRR-2 lays in the region of weak spatial field distribution. Hence, it requires larger structural asymmetries (higher threshold) to excite the Fano resonance, which also results in the observed relatively higher $Q$-factors for a specific value of $\delta$, as shown in Fig. 3c, d. Whereas, while decreasing the asymmetry by actuating SRR-2 cantilevers, the SRR-1 that is positioned on the substrate is in the strong near-field spatial region and is likely to possess greater influence on the near-field coupling between the resonators and hence results in a stronger excitation of Fano features possessing relatively lower $Q$-factors (lower threshold). Hence, the resulting MIO states in the electro-optical characteristics of the proposed MEMS Fano-metasurface fundamentally exploits the intrinsically present exponentially decaying spatial distribution of near-fields extended along the $z$-axis of the sample.

One of the important aspects of MEMS-based metasurface device is that it could be operated in both volatile as well as in non-volatile regimes depending on the partial or complete actuation of the constituent resonator cantilevers in the unit cell. This directly reflects on the persistent repeatability of the demonstrated logic gate functionalities of MEMS Fano-metasurface. As emphasized earlier, the results on XOR logic operation of the proposed device discussed in Fig. 4a–d represent non-volatile operation regime of the MEMS Fano-metasurface, where due to stiction in the MEMS devices, the cantilevers remain stuck to the substrate even after the input voltage is removed (refer to Supplementary Movie 1). This non-volatile property of the device affects the speed and repeatability of the device operation, but enables the memory features in the device, which could potentially be used as memory registers in data storage and processing techniques[33]. On the other hand, the volatile feature of MEMS Fano-metasurface enabled by the partial actuation of SRR-1 and SRR-2 cantilevers (applied voltage of $V_{1,2} < 25$ V, i.e. less than the pull-in voltage of the device) assures persistent repeatability of the XOR and other logic operations (refer to Supplementary Movie 2). Most interestingly, the volatile feature of XOR functionality in the device possesses unique property of pseudorandom generation and serves as a key component in OTP encryption/decryption techniques in establishing theoretically secured cryptographic protocols in the communication systems. In Fig. 4e, we provide a schematic for the secured OTP cryptographic channel that can be achieved by performing the XOR logical operations using the MEMS Fano-metasurface in the volatile operation regime of the cantilevers. The input/output states of the XOR logic operation are listed by the truth table shown in Fig. 4. The two inputs (secret key) $k = \{0,1\}^n$ and (private message) $m = \{0,1\}^n$ are expressed as the strings of binary digits that represent the structural states of resonators SRR-1 ($V_1$) and SRR-2 ($V_2$). The transmitted optical (THz) bits are represented by $F$ that signifies absence (binary '$0$') or presence (binary '$1$') of Fano intensity states. At the source (Alice's end), for each bits of $k = (1010)$ and $m = (1100)$, a XOR operation ($F = k \oplus m$) is carried out by performing an optical readout using THz beam that encrypts the private message ($m$) in the form of transmitted optical bit, $F = (0110)$, which is sent as a ciphertext (encrypted message) through the unsecured public channel. At the destination (Bob's end), the optical bits (ciphertext) are detected using a photo-detector (PD) and the resulting voltage states ($F = 0110$) from the PD are directly fed through a trigger channel to the voltage source $V_2$ that controls the actuation states $\{0,1\}$ of SRR-2 of MEMS Fano-metasurface. Finally, the ciphertext ($F$) containing the information of private message is decrypted to retrieve original private message ($m$) at the destination end by performing the inverse XOR operation, $m = k \oplus F$, as shown in Fig. 4e. The secret OTP key ($k$) contains the information of the structural state of SRR-1 ($V_1$) and is pre-shared between the source and the destination ends via a private (secure) channel. Thus, the proposed far-field XOR functionality of the MEMS Fano-metasurface could open-up new avenues for realizing cryptographically secured wireless THz communications[44,45].

In summary, we demonstrated excitation of sharp Fano resonances in a reconfigurable MEMS metasurface using two independent voltage controls that constitutes digital XOR, XNOR, NOT, NAND and OR logic gates at THz frequencies. Formation of MIO states resembling the hysteresis loop is observed in the electro-optical properties of the MEMS Fano-metasurface that results from the anisotropic variation in the near-field coupling of Fano resonance excitation during increasing and decreasing out-of-plane asymmetry of the system. The XOR operation of the device reveals that the concept can show potential prospects in super-encryption techniques in *i*-banking sectors, short

messaging services (SMS), defence, national data security systems and high speed cryptographically secured wireless communication networks, which are now being pushed towards THz frequencies. The NAND logic operation being the universal logic function would enable the construction of all the other Boolean logic operations, thereby providing a flexibility of enhancing the digital functionalities of the device. The reported multifunctionalities of the proposed MEMS Fano-metasurface are largely suitable for real-world applications such as active sensors possessing tuneable mode volumes, nonlinear devices and modulators. Alongside, the MIO characteristics of the MEMS Fano-metasurface could potentially provide a flexible platform for developing the next generation randomly accessible, digital and programmable metamaterials for precise tailoring of electro-optical properties and multichannel data processing at higher bit rates.

## Methods

**Sample fabrication.** The MEMS Fano-metasurface was fabricated using a CMOS compatible process as described below. First, the lightly doped 8 in. silicon substrate of 725 μm thickness was cleaned and a 100 nm thick sacrificial $SiO_2$ layer was deposited using low pressure chemical vapor deposition (LPCVD) process. Following this, conventional photolithography process was used to pattern the anchor region. With the designed pattern, the parts of sacrificial $SiO_2$ for anchor regions were dry etched using reactive ion etching process. After this, a 50 nm thick $Al_2O_3$ layer was deposited using the ALD process, followed by the sputter deposition of Al metal of thicknesses 300, 500, 700 and 900 nm. Note that the bimorph layers (Al/$Al_2O_3$) were in physical contact with Si substrate at the anchor region, and in the remaining part of the wafer, it was on top of the sacrificial $SiO_2$ layer. Then, the second photolithography step was carried out for defining the cantilevers and metal lines of metasurface patterns. Following this, both Al and $Al_2O_3$ layers were dry etched to form the designed metasurface. Finally, vapor hydrofluoric acid (VHF) was used to isotropically etch the $SiO_2$ sacrificial layer underneath the bimorph structures, thereby suspending it over the Si substrate with an air gap between them. At the anchor region, since the bimorphs were in physical contact with Si substrate; the VHF release process was not time controlled, and this ensured higher yield of the devices. Due to the residual stress in the bimorph cantilevers, the released cantilevers were bent up, thereby increasing the initial tip displacement.

**Electromechanical characterization of the MEMS device.** The deflection/actuation profiles of released cantilevers were measured using Lyncee Tec. reflection digital holographic microscope (R-DHM). The released chips are wire-bonded to a printed circuit board (PCB). Separate voltage supplies ($V_1$ and $V_2$) are used for the actuation of SRR-1 and SRR-2 cantilevers, respectively. Silicon (Si) substrate was chosen as the ground potential, and the cantilevers were positively biased. When voltage is applied across the released cantilevers and Si substrate, the attractive electrostatic force deforms the suspended cantilevers towards the fixed Si substrate. This mechanical deformation of cantilevers induces a restoring force that opposes the electrostatic force causing the deflection at the first place. Hence, the final position of the cantilever at a given voltage is determined by the equilibrium position, where the electrostatic force and restoring force balances each other. As the applied voltage increases, the electrostatic force exceeds the restoring force and at a critical value known as the pull-in voltage (>25 V), thereby bringing the cantilevers to be in physical contact with Si substrate (shown in the inset of Fig. 2b) (refer to Supplementary Movie 1). The pull-in can be clearly observed through the optical microscope fitted on the R-DHM. The $Al_2O_3$ layer beneath the Al layer ensured that there is no current flowing from Al layer to Si substrate, when pull-in occurs. Current flow through the Al/Si junction is forbidden as that could permanently damage the device due to the increase in the local temperature.

**THz measurements.** The MEMS Fano-metasurface is optically characterized using the conventional GaAs photoconductive switch-based THz-time domain spectroscopy system operating in the transmission mode. The wire-bonded MEMS metasurface sample is positioned at the focus of the THz beam. The electrical connections to the SRR-1 and SRR-2 resonators structures are established using a two channel DC voltage source. For four configurations of the voltages ($V_1 = 0$ V, $V_2 = 0$ V; $V_1 = 35$ V, $V_2 = 0$ V; $V_1 = 0$ V, $V_2 = 35$ V; and $V_1 = 35$ V, $V_2 = 35$ V), the THz wave of beam spot 4 mm impinges on the sample at normal incidence and the transmitted THz pulse is captured using the THz detector connected to the lock-in amplifier. THz response through the bare silicon substrate is measured as the reference. In the post-processing steps, the detected THz pulses measured through the sample and the bare substrate are fast Fourier transformed (FFT) to obtain the corresponding THz spectra. Later, the transmitted THz spectrum through the sample ($T_S(\omega)$) is normalized with respect to the transmission through the substrate ($T_R(\omega)$), i.e. $T(\omega) = |T_S(\omega)/T_R(\omega)|$, and the normalized spectrum is shown in Figs. 2a, b and 4a–d.

**Numerical simulations.** FDTD numerical simulations were conducted to calculate the THz transmission spectra and the confined electric near-fields and surface current distributions corresponding to the resonance modes for the normal incident of THz waves of TE polarization. Full-field electromagnetic wave simulations were performed using the commercial simulation software CST microwave studio. For the material property, aluminium (Al) of thickness 900 nm was modelled as a lossy metal with a conductivity of 3.57e7 S/m. Aluminium oxide and Silicon were modelled as lossless dielectric materials with a dielectric constant of 9.5 and 11.9, respectively. In the simulation, a single unit cell of the metasurface structure was simulated with periodic boundary conditions employed in axial directions orthogonal to the incident waves. The perfectly matched layers are applied along the propagation of the electromagnetic waves. Plane waves were incident onto the unit cell from the port on the metal side, while the transmission spectrum was determined from the probe placed at the other side of metasurface. The experimentally measured (inset in Fig. 2b) deformation (released) angles for the cantilevers establishes the congruence between the values of applied voltages in the measurements and the structural asymmetry parameter used in the simulations. For the numerical simulations, the deformation angle of the cantilevers is calculated using the expression, $\theta = \sin^{-1}(h/s)$, where $h$ is the released height of cantilevers and $s$ is the length of cantilever. In the meanwhile, field monitors are used to collect the electric fields, magnetic fields and the respective surface currents at Fano resonance frequencies for varying asymmetry values.

**Code availability.** The transmission responses and the electric field distribution plots were numerically computed using CST microwave studio.

## Data availability

The data that support the findings of this study could be made available upon request to the corresponding author. The data from this paper is also available from the University of Southampton ePrints research repository: https://doi.org/10.5258/SOTON/D0612

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

## Acknowledgements

M.M., P.P. and R.S. acknowledge the funding support from Ministry of Education, Singapore, MOE2017-T2-1-110 (S) grants and National Research Foundation (NRF) Singapore and Agence Nationale de la Recherche (ANR), France-NRF2016-NRF-ANR004 (M4197003) grant. M.M., P.P., N.I.Z. and R.S. acknowledge the funding support from Ministry of Education, Singapore, MOE2016-T3-1-006 grant. N.I.Z. acknowledges the support from the UK's Engineering and Physical Sciences Research Council (grant number EP/M009122/1). C.L. acknowledges the grant support from NRFCRP15-2015-02 "Piezoelectric Photonics Using CMOS Compatible AlN Technology for Enabling the Next Generation Photonics ICs and Nanosensors" at NUS, Singapore.

## Author contributions

M.M., R.S. and P.P. conceived the idea and designed the experiments. P.P. and N.W. fabricated the experimental samples. M.M. carried out the THz measurements and numerical simulations. M.M., P.P., N.I.Z., N.S., C.L. and R.S. analysed the results. M.M. and R.S. prepared the manuscript with inputs from the co-authors. R.S. supervised the overall project.

## Additional information

**Competing interests:** The authors declare no competing interests.

