## [Peer Review File · Nature Communications]

Reviewers' comments:

Reviewer #1 (Remarks to the Author):

This manuscript reports on XOR, NOT, and NAND operation using a MEMS reconfigurable metamaterial exhibiting multiple-input-output (MIO) characteristics in its near- and far-field optical properties. The authors smartly utilized the electric-field-induced structural asymmetry of the structure and its associated Fano resonance.

Although the present study by M. Manjappa et al., is actually not the first experimental demonstration of MIO XOR operation as claimed by the authors (See, for example, XOR, AND, OR, and 2-bit DAC operations in electrically-controlled MIO graphene/ferroelectric nonvolatile metasurfaces, W. Y. Kim et al., Nature Communications, 7, 10429, 2016), the results reported here are of high-impact and will be of interest to broad photonics and metamaterials research communities as the efficiency (or contrast) of the signal significantly is improved over the past research. Therefore, I recommend publication in Nature Communications as I believe this work will open many other opportunities and activities in MEMS-actuated logical metasurfaces.

In reading the manuscript, a few minor questions came up.

(1) The NOT gate operation seems to be a trivial case because any metasurface (or metamaterial) exhibiting an external-stimulus-driven transmission change with a proper threshold setting. Please explain why the NOT gate operation demonstrated here deserves a special attention.

(2) Aren't there any latching problem of the cantilever arms with the application of the voltages? Please show a stable sequential operation with multiple pulses of electrical signals in addition to the results only with a static application of the voltages in a cycle.

(3) For the NAND operation, a near-field reading will be required to obtain the logical operation result. This is neither an easy task, nor a practical one. The authors need to address this issue in the revised manuscript.

Reviewer #2 (Remarks to the Author):

The authors have demonstrated experimentally and theoretically a MEMS-actuated metamaterial (or metasurface) with 2 electrical inputs connecting parallel to each side of SRR meta-atoms. This can exhibit XOR and NOT output by taking far-field characteristics (i.e. difference of transmittances), and NAND output by taking near-field characteristics (i.e. SNOM-like measurement) based on the structural tunability of the asymmetry. The device has been fabricated accurately and showed well-organized behaviors corresponding to the simulations, which are beautiful. However, after the reviewer's consideration, this work does not have a high enough impact for publication in Nature Communications because;

- 1) The reviewer does not believe reading out the typical logic operations by using THz pulses or near-field microscopy would be versatile
- 2) MEMS-actuated metamaterials with two different channels in THz regime has been already demonstrated by other group (<http://dx.doi.org/10.1063/1.4944711>)
- 3) It was not clear how to compose other logic gates based on the suggested near-field NAND gate (hard to imagine the cascade connection of them)

Other questions & comments

1. Maybe it is too few experimental plots in Figs. 3(a) and 4(a) (the each number of the point is exactly corresponding to the number of the curves in Figs. 2(a) and (b)). If it is possible, the

authors should add more experimental points.

2. The reviewer felt the explanations for NOT operation (Fig. 4) can be briefed since it is included in XOR table with fixing either $V_1 = "1"$ or $V_2 = "1"$.

3. By referring supplementary, the reviewer could understand the Al thickness can tune the released height. However, still it was not clear what the exact reason of choosing 900 nm-thick Al for the actual fabrication.

4. Is the multiple input more than 3 or 4 possible? If we can increase the number of the input in the future, that would be very interesting and more functional.

5. Some characters are too small in Figs. 3(c) and (d).

Response Letter to the Reviewers

We thank the reviewers for their time and precise reading of the manuscript and for their valuable comments and suggestions. All the comments from the reviewers have been answered with greater attention and details and the same is highlighted in blue font. The revisions in the manuscript are highlighted with yellow background.

Reviewers' comments:

Reviewer #1 (Remarks to the Author):

This manuscript reports on XOR, NOT, and NAND operation using a MEMS reconfigurable metamaterial exhibiting multiple-input-output (MIO) characteristics in its near- and far-field optical properties. The authors smartly utilized the electric-field-induced structural asymmetry of the structure and its associated Fano resonance. Although the present study by M. Manjappa et al., is actually not the first experimental demonstration of MIO XOR operation as claimed by the authors (See, for example, XOR, AND, OR, and 2-bit DAC operations in electrically-controlled MIO graphene/ferroelectric nonvolatile metasurfaces, W. Y. Kim et al., *Nature Communications*, 7, 10429, 2016), the results reported here are of high-impact and will be of interest to broad photonics and metamaterials research communities as the efficiency (or contrast) of the signal significantly is improved over the past research. Therefore, I recommend publication in *Nature Communications* as I believe this work will open many other opportunities and activities in MEMS-actuated logical metasurfaces.

Answer:

We thank the reviewer for his/her precise analysis of the presented results and for appreciating the idea and the concept realized in this work. We sincerely acknowledge reviewer's precise understanding of our work and his/her views on the novelty and quality of our results. We greatly thank the reviewer for the suggestions, comments and recommendation of the manuscript for publication at *Nature Communications*.

We apologize for not noticing the relevant work highlighted by the reviewer. We have compared our studies with the results discussed in the cited reference ^[1], and would like to

highlight the critical difference in the observed MIO (hysteresis-type) features in both the studies. In addition to reviewer's observation of the studies in terms of better efficiency i.e. contrast in the output states, our results also highlight on a different origin for the observed MIO states in the proposed MEMS Fano-metamaterial system. In the cited work ^[1] the MIO states result from the reversal of polarization of the ferroelectric polymer and hence the overall performance of the device is limited by the material properties of the ferroelectric layer, whereas in our studies the MIO (hysteresis type of loop) states are created from exploiting the anisotropic near-field coupling in the system, which is based on the structural three dimensional geometry of resonators and doesn't involve any change in the material properties. Since our results are purely metamaterial enacted MIO states, it could be extended from THz to optical frequencies using the MEMS/NEMS metamaterial designs. We greatly thank the reviewer for his/her insights and for bringing up the much-related work ^[1] to our notice. In the revised manuscript, we have cited the work appropriately highlighting on the distinctive nature of the results and the physical mechanisms, along with the enhanced performance. Please refer to the changes in the manuscript on page 3, paragraph 1.

In reading the manuscript, a few minor questions came up.

(1) The NOT gate operation seems to be a trivial case because any metasurface (or metamaterial) exhibiting an external-stimulus-driven transmission change with a proper threshold setting. Please explain why the NOT gate operation demonstrated here deserves a special attention.

Answer:

We agree with the point highlighted by the reviewer that the NOT operation is a trivial case in many of the active metamaterial system that uses the external field to switch ON and OFF the resonance features. However, the NOT logical operation presented in the manuscript is a special case of the XOR function for one of the structural state being snapped down on the substrate using the voltage source. The presented NOT operation is represented as the result of formation/annihilation of a new resonance mode (Fano mode) in the absence/presence of external stimulus in the coupled metamaterial system. This negation operation in our design represents the switching between the coupled and uncoupled regime of the near-field interactions, where it signifies the negation property in the Fano resonance feature in the

presence of external stimulus. Here, we demonstrate the system exhibiting the XOR functionality that can also be used as the negation (NOT) operator by closing the input possessing high ('1') logic state and other input being varied. This condition leads to the inverter or the NOT gate possessing high contrast output intensity. On the other hand, by closing the input having the low ('0') logical state, the buffer/PASS gate can be realized, where the operation does not alter the input state and output logic state is same as the input logic state. Hence, based on the state of one of the SRRs, the control of other SRR leads to a NOT or PASS logic operation, and hence this feature shows the pre-selection of operation between NOT to PASS gating logic through the added control signal. Please refer to the changes in the manuscript on page 9. Paragraph 2.

To remove this ambiguity, we move the figure representing the NOT gate from the manuscript to the supplementary information (Fig. S14) and in the revised manuscript we have provided a more detailed analysis and implications of non-trivial logical operations such as XOR, XNOR, NAND and OR functions.

(2) Aren't there any latching problem of the cantilever arms with the application of the voltages? Please show a stable sequential operation with multiple pulses of electrical signals in addition to the results only with a static application of the voltages in a cycle.

Answer:

We thank the reviewer for this important question. In practical applications MEMS devices suffer from the limitations in the form of stiction, which prevents the device in enduring repeatable operations. Due to stiction, the cantilevers that are being snapped down using the voltages do not relax back to the initial state (they stick to the surface) under the removal of voltage input, which results in the nonvolatile nature of the device operation. This has been the major drawback of MEMS/NEMS devices and is an on-going topic of research in the MEMS community, which has seen a good progress by proposing several methods to overcome the stiction, where one of the ways is by using the nematic liquid crystals^[2], surface modifications or engineering the micro-actuator design. .

Usually for our previous works, we do a special temperature treatment just before and after the vapor hydrofluoric acid (VHF) release process that allows for the repeatable operation. **[Redacted]**. In our current devices, we could not do such treatment owing to the safety concerns in the

cleanroom. Hence, we observe latching of some of the microcantilevers, when operated at room temperature in ambient conditions. *However, we would like to highlight the fact that our design can allow the excitation of Fano resonances without the cantilevers being completely snapped down on the substrate and hence the problem of stiction/latching (absolute voltage, V_{DC} is less than the pull-in voltage of the device) can be easily circumvented.* Hence, practically, the stable sequential operation of the device is possible and is subject to the stability and quality of the fabricated samples. In the manuscript, we have clearly highlighted on this aspect of the device and described our measurement processes to remove the possible ambiguity on the sequential operation (on page 8, paragraph 1 of the manuscript).

(3) For the NAND operation, a near-field reading will be required to obtain the logical operation result. This is neither an easy task, nor a practical one. The authors need to address this issue in the revised manuscript.

Answer:

We thank the reviewer for bringing his/her concerns on the near-field aspect of the NAND operation in the currently proposed design. The tip based THz near-field microscopy^[3-5] can be used as one of the ways to retrieve the near-field information in terms of output field intensity states that in-turn can be used to sequentially trigger the input voltage of the cascaded NAND device. This could enable the near-field cascading of the logic gates in the CMOS configuration. We would like to clarify here that the experimental demonstration of the near-field detection of the NAND operation is beyond our experimental capabilities, however, the well-established near-field tip based THz probes can assist in practically realizing the NAND operation in the proposed MEMS Fano metamaterial design.

Further, to address the limitations of the near-field entity of NAND functionality of the device, we conceptually show the NAND logical operations in the far-field intensity spectrum using the same geometrical design and concepts of MEMS Fano-metamaterial but by combining two unit-cells to make one composite unit-cell (super-cell) consisting of two Fano meta-molecules (a pair of SRR-1 - SRR-2 and SRR-3 - SRR-4), as shown in the Fig. R1 (a). By independently controlling the SRRs present within the two Fano meta-molecules using the voltage sources (V_1 and V_2) the NAND logical function is realized by the THz readout pulse in the form of presence and absence of Fano resonance in the far-field intensity spectrum. As shown in the Fig. R1 (a) below, for realizing NAND functionality, the initial

configuration of the super-cell of the device is designed such that one of the two resonators in the Fano meta-molecule (say, SRR-1 and SRR-3) are attached on the substrate and other two resonators (say, SRR-2 and SRR-4) are in the released states and are connected to voltage controls (V_1 and V_2) as two logical inputs. The geometrical dimensions of the metamolecule resonators in the super-cell are same as the MEMS Fano-metamaterial design discussed in the manuscript, while the periodicity of the structure along the x -axis is doubled ($p_x = 220 \mu\text{m}$ & $p_y = 75 \mu\text{m}$) to form the super-cell consisting of a pair of Fano metamolecules. The far-field transmission spectra of the device are shown in Fig. R1 (b), where the transmission spectrum depicted in (i) shows the strong excitation of Fano resonance feature at 0.53 THz observed in the far-field for no voltage inputs ($V_1 = 0$ & $V_2 = 0$), that results in the true output state of the device ($F = 1$). Similarly, for the device configurations with the voltage applied either on SRR-2 or on SRR-4 ($V_1 = 1$ and $V_2 = 0$ / $V_1 = 0$ and $V_2 = 1$), there is weak excitation of the Fano resonance at 0.53 THz, which signifies the true output state ($F = 1$). The reduced resonance strength in Fano resonance is caused due to the 50% reduction in the number of resonant Fano meta-molecules. For the last configuration, when both voltages ($V_1 = 1$ & $V_2 = 1$) are applied on the SRR-2 and SRR-4 of metamolecules, the Fano resonance features completely disappears, thereby resulting in the false output state ($F = 0$) of the optical readout. Thus, the proposed electro-optical feature of the MEMS Fano metamaterial forms the NAND logical operation, with possessing the true output states when either one/both input states are false. Hence, these intriguing features shown in the near-field and the far-field signatures greatly merit the multifunctional aspect of metamaterials and can lay a useful platform in realizing the digital and programmable metamaterials possessing high efficiency and high output contrast ratios.

More interestingly, if the states of SRR-1 and SRR-3 are kept in the OFF state (released) instead of ON state (snapped down), the OR logical operation can be realized based on the control inputs to SRR-2 and SRR-4 as shown in Figure R2. Thus, by using the same configuration of the device but with complementary control states, we can realize both NAND and OR logical outputs in the far-field amplitude/intensity states of the device. We have included a small description of this concept on NAND and OR logic gates in our revised manuscript on page 10 paragraph 2.

Figure R1. (a) Artistic diagram showing the design of the NAND logical metamaterial structure with the highlighted unit cell consisting of four symmetric SRR components (SRR-1, SRR-2, SRR-3 and SRR-4) forming a super-cell. SRR-1 and SRR-3 are attached on the substrate, whereas the resonators SRR-2 and SRR-4 are initially in the released state and actuated using the voltage inputs V_1 and V_2 . (b) Showing the numerically simulated spectrum for the various (00, 10, 01 and 11) structural input configurations of the metamaterial possessing the true ($F = 1$) and false ($F = 0$) output states in the form of presence and absence of Fano resonance feature at 0.53 THz of the transmission spectrum. In the inset, the truth table of the metamaterial operation is listed for NAND logical operation.

Figure R2. Representing the numerically simulated transmission spectrum of the metamaterial structures depicted in the inset of the respective plots. The electro-optical operation of the device results in the OR truth table for the input states of $V_1V_2 = 00, 10, 01$ and 11 giving the outputs as $F = 0, 1, 1$ and 1 respectively, in the form of true ($F = 1$) and false ($F = 0$) states of Fano resonance feature in the far-field amplitude spectrum.

Reviewer #2 (Remarks to the Author):

The authors have demonstrated experimentally and theoretically a MEMS-actuated metamaterial (or metasurface) with 2 electrical inputs connecting parallel to each side of SRR meta-atoms. This can exhibit XOR and NOT output by taking far-field characteristics (i.e. difference of transmittances), and NAND output by taking near-field characteristics (i.e. SNOM-like measurement) based on the structural tunability of the asymmetry. The device has been fabricated accurately and showed well-organized behaviors corresponding to the simulations, which are beautiful.

Answer:

We sincerely thank the reviewer for his/her remarks on the results and for applauding the consistency of the data presented in the manuscript. We are thankful for all the constructive critiques and comments from the reviewer and we have responded to each of them by providing additional information and have applied our best efforts to address the reviewer's following concerns.

However, after the reviewer's consideration, this work does not have a high enough impact for publication in Nature Communications because;

Answer:

Whilst, we welcome the reviewer's critiques and comments, we respectfully disagree with his/her opinion on the novelty and impact of the results to meet the standard of *Nature communications*. In the following, we highlight on the key features of our results that emphasize the novelty and impact of our work in comparison with the earlier demonstrations in the broad field of active photonic devices.

1. For the first time, an anisotropic near-field coupling in the metamaterial/plasmonic system that leads to the multiple-input-output (MIO) features is demonstrated, which shows a hysteresis type-closed loop behavior in the near- as well as far-field resonance electro-optical characteristics of the metamaterial device, which could lead to both volatile and nonvolatile memory effects in system.
2. First demonstration of excitation and active control of sharp Fano-resonance features in a MEMS metamaterial structure due to the nanometric changes ($\geq \lambda/15000$) in the structural asymmetries. Such Fano features give rise to a strong field confinement in a

tight volume that can be used in plasmonic sensors applications possessing actively tunable mode volumes for refractive index sensing and biosensing applications. Further, the sensitive nature of the microscopic system to a nanometric (deep-subwavelength) change in the asymmetry of the structure can be a new regime of exploring tera-nano photonics.

3. The device configuration that we have proposed has a unique advantage of probing the near-fields in the 3rd dimension of the sample that exploits the nonlinear (exponential) decay of the near-field (evanescent) waves in the aerial direction of the sample. To the best of our knowledge, this anisotropic coupling feature of near-field via the Fano excitation in the aerial dimension wasn't probed before and the realization of metamaterial enacted MIO states could lead to many fascinating phenomena in the field of reconfigurable and functional metamaterials and could lay a reliable platform for developing the digital and programmable plasmonic devices.
4. We present a versatile functional metamaterial exhibiting versatile and nontrivial logical operations, such as XOR, XNOR, NAND, OR logical operations in the far-field and near-field optical characteristics possessing high contrast output states. More importantly, XOR and XNOR logical functions are not linearly separable into basic logic gates and hence they are difficult to realize in the photonic system. More importantly, the independent control of tightly coupled resonator system allows for the selection between even a NOT/PASS or NAND/OR logical operation. This functionality of selection of logic gate operation is truly unique for our system. Further, we highlight on the possible applications of the XOR logical operation in achieving cryptographically secured classical information channel in the sub-terahertz bandwidths by sharing the secured key in the private domain. This cryptographic feature of our device is highlighted in the revised manuscript and explained in more detail in the supplementary information. Further, the versatility of the device in realizing multiple logical operations can lay a perfect platform for realizing digital and programmable metamaterials.
5. Since the terahertz part of the spectrum lies in between the electronics and photonic part of the electromagnetic spectrum, there is a huge lack of efficient devices in digitizing the electro-optical properties across the THz spectrum. Hence, metamaterials or graphene based^[1] devices enabling many intriguing properties at THz frequencies could lay a platform for realizing digital or programmable metamaterial at these frequencies. Hybrid materials-graphene interface based logic

gates at THz showed low efficiency in the output signals and are purely material and frequency dependent^[1]. Hence, our MEMS based metamaterial design showing versatile logic gates such as OR, NAND, XOR, XNOR and NOT operations could be the reliable and more efficient devices in performing the digital functionalities at sub-terahertz and terahertz frequencies and the concept can also be extended to near-infrared and optical frequencies using the NEMS designs.

1) The reviewer does not believe reading out the typical logic operations by using THz pulses or near-field microscopy would be versatile

Answer:

We thank the reviewer for bringing up this point about the readout and versatility of the near-field observed NAND logical operations. We absolutely agree that in the metamaterial/plasmonic research community retrieving the information using the near-field probe is challenging. However, recent demonstrations (like SNOM techniques as suggested by the reviewer) mainly in the THz part of the electromagnetic spectrum have successfully shown the readout of the near-field data using the tip based near-field microscopy^[3-5]. In the revised manuscript, we have tried to address this concern and show that the NAND operation can also be realized in the far-field regime of the device operation, as depicted in Fig. R3. The configuration of the NAND device slightly varies in comparison with the proposed XOR device, where the unit cell consists of four SRR resonators forming a super-cell consisting of a pair of Fano metamolecules (a pair of SRR-1 - SRR-2 forming metamolecule-1 and SRR-3 - SRR-4 forming metamolecule-2). Geometrical dimensions of the metamolecule resonators in the super-cell are same as the MEMS Fano-metamaterial design discussed in the manuscript, while the periodicity of the structure along the x -axis is doubled ($p_x = 220 \mu\text{m}$ & $p_y = 75 \mu\text{m}$) to form the super-cell comprising of a pair of metamolecules. By independently controlling the SRRs present within the two metamolecules using the voltage sources (V_1 and V_2), the NAND logical function is realized by the THz readout pulse in the form of presence and absence of Fano resonance in the far-field amplitude/intensity spectrum. As shown in the Fig. R3 (a) below, for realizing NAND functionality, the initial configuration of the super-cell of the device is designed such that one of the two resonators in each of the meta-molecule (say, SRR-1 and SRR-3) are attached on the substrate and other two resonators (say, SRR-2

and SRR-4) are in the released state and are connected to voltage controls (V_1 and V_2) as two logical inputs.

Figure R3. (a) Artistic diagram showing the design of the NAND and OR logical metamaterial structure with the highlighted unit cell consisting of four symmetric SRR components (SRR-1, SRR-2, SRR-3 and SRR-4) forming a super-cell. SRR-1 and SRR-3 are attached on the substrate, whereas the resonators SRR-2 and SRR-4 are initially in the released state and actuated using the voltage inputs V_1 and V_2 . (b) Showing the numerically simulated spectrum for the various (00, 10, 01 and 11) structural input configuration of the metamaterial showing the true ($F = 1$) and false ($F = 0$) output states in the form of presence and absence of Fano resonance feature at 0.53 THz of the transmission spectrum. In the inset, the truth table of the metamaterial operation is listed for NAND logical operation.

Figure R4. Depicts the numerically simulated transmission spectrum of the OR metamaterial depicted in the inset of the respective plots. The electro-optical operation of the device results in the OR truth table for the input states of $V_1V_2 = 00, 10, 01$ and 11 giving the outputs as $F = 0, 1, 1$ and 1 respectively, in the form of true ($F = 1$) and false ($F = 0$) states of Fano resonance feature in the far-field amplitude spectrum.

The far-field transmission spectra of the device are shown in Fig. R3 (b), where the transmission spectrum depicted in (i) shows the strong excitation of Fano resonance feature at 0.53 THz observed in the far-field for no voltage inputs ($V_1 = 0$ & $V_2 = 0$), that results in the true output state of the device ($F = 1$). Similarly, for the device configurations with the voltage applied on either SRR-2 or SRR-4 ($V_1 = 1$ and $V_2 = 0$ / $V_1 = 0$ and $V_2 = 1$), there is a weak excitation of the Fano resonance at 0.53 THz, which signifies the true output state ($F = 1$). The reduced resonance strength in Fano resonance is caused due to the 50% reduction in the number of resonant Fano meta-molecules. For the last configuration, when both voltages ($V_1 = 1$ & $V_2 = 1$) are applied on the SRR-2 and SRR-4 of metamolecules, the Fano resonance features completely disappears, thereby resulting in the false output state of the optical readout. Thus, the proposed electro-optical feature of the MEMS Fano metamaterial forms the NAND logical operation, with possessing the true output intensity states when either one/both input voltage states are false. Hence, these intriguing features shown in the

near-field and the far-field signatures lay a useful platform in realizing the functional and programmable metamaterials possessing high efficiency and high output contrast ratios. In Fig. R4, we also show the realization of OR logical function using the proposed MEMS Fano- metamaterial design in the far-field transmission amplitude spectrum by changing the initial state of SRR1 and SRR3 to be in released state, which is a complementary to the NAND metamaterial structures. We have included this new data and figures in the Supplementary Information (Fig. S15 and S16) and have given a small description on NAND and OR functionality of the device on page 10 paragraph 2 of the revised manuscript file.

2) MEMS-actuated metamaterials with two different channels in THz regime has been already demonstrated by other group (<http://dx.doi.org/10.1063/1.4944711>)

Answer:

We thank the reviewer for citing this work. However, we respectfully disagree and would like to emphasize that our work is different and novel both in terms of design and concepts compared to the cited article^[6]. The basic geometrical design and the operation configuration of the device discussed in our work and the work described in the article cited by the reviewer are very different in several ways as described below:

1. In cited article, the optical properties are mainly altered by modifying the optoelectronic properties of the semiconducting material via voltage biasing, thereby modulating the intensity of the incoming THz wave. Whereas, in our proposed MEMS design, we alter the electro-optical properties of the device by purely modifying the structural/geometrical parameters of the metamaterial structure, which exploits the intriguing feature of near-field coupling in the metamaterial system.
2. Our proposed MEMS Fano-metamaterial design and the results are remarkable that aids the multi-functionalities of the device in terms of demonstrating the logic operations by digitizing the input and output states by creating the multiple-input-output configuration in the electro-optical properties of the system. Whereas, the results shown in ref [6] totally different and focuses on the THz amplitude modulation of the individual resonances of the two unit-cell designs in the complementary metamaterial structures connected to voltage controls in the Schottky configuration.
3. Although there exist few designs involving the multiple controls in the recent times, however, our results are one of the very few demonstrations to use the multiple input

controls within the unit-cell of strongly coupled metamolecule and to reconfigure the optical properties of the metamaterial by altering the strong near-field coupling in all the three dimensions of the sample.

4. We further claim that our results are the first demonstration of creating the intriguing feature of multiple-input-output states by introducing anisotropic nature of purely metamaterial enacted strong near-field coupling features that lead to memory effects, enhanced electro-optical performance with high contrast ratios and could aid in realizing optical properties on demand using plasmonic/metamaterial systems.

With all these justification, however we think that the reference is related to the work in terms of using two electrical actuations for modulating the THz waves. Hence, we cite this article in the revised manuscript (Ref. 18 on page 2, paragraph 1 of the revised manuscript).

3) It was not clear how to compose other logic gates based on the suggested near-field NAND gate (hard to imagine the cascade connection of them)

Answer:

We thank the reviewer for this important question. We agree with the reviewer that near-field aspect of the NAND operation makes the cascading of the system non-trivial. The tip based THz near-field microscopy^[3-5] can be used as one of the ways to retrieve the near-field information in terms of output field intensity states that in-turn can be used to sequentially trigger the input voltage of the cascaded NAND device. This could enable the near-field cascading of the logic gates in the CMOS configuration. *However, we also address this issue by proposing the NAND functionality of the device in the far-field transmission spectra that would facilitate the cascading aspect of the NAND operation to realize other logic gates, as shown in Fig. R5.* In the Supplementary Information, we provide a realistic schematic for achieving cascading of the far-field NAND metamaterial devices to form OR and AND logical outputs. The cascading of the gates would work similar to the electronic counterparts. However, the proposed cascading of NAND metamaterial is unlike the conventional transistor or waveguide based cascading of gates, where either solely electrical or optical channels are used to realize the cascading networks. In the current device, the optical readout from the metamaterial is converted to electrical signals using the standard photodetectors and used as the trigger to the electrical inputs of the cascaded NAND metamaterial. In the

following Fig. R5, we show the extraction of OR and AND logical functionalities by cascading the NAND metamaterials operation in the far-field.

We have included this aspect and the practicality of cascading the NAND metamaterial gates to realize other logical operations such as OR/AND functionalities in the revised manuscript as well as in the Supplementary Information Fig. S17. We believe that the demonstration of the versatile logical operations and the multiple-input-output states in the proposed design of MEMS metamaterials could serve as the platform for realizing multifunctional metamaterials. Further, the realization of the NAND operation both in the near- and the far-field optical properties can provide a versatility of developing both near-field and far-field operation of programmable metamaterials.

Figure R5. (a) and (b) Schematic showing the concept and design of cascading the NAND metamaterials operating in the far field to realize the OR and AND logical functionalities.

Other questions & comments

1. Maybe it is too few experimental plots in Figs. 3(a) and 4(a) (the each number of the point is exactly corresponding to the number of the curves in Figs. 2(a) and (b)). If it is possible, the authors should add more experimental points.

Answer:

We thank the reviewer for this precious suggestion. We have performed additional rigorous measurements to obtain more data and have revised the figures 2 (a) and 2 (b) of manuscript

by providing more curves in the experimental plots. We also revised the Fig. 3 representing the multiple-input-output (MIO) characteristics with more number of experimental data points. The MIO curve is plotted with respect to the varying differential voltage defined as modulus of difference between the two input voltages V_1 and V_2 represented by $\Delta V = |V_1 - V_2|$. Furthermore, we provide new results showing the multiple-input-output configuration for the measured Q -factors of the Fano resonance curves that lead to the realization of XNOR logic operation in the far-field Q -factor (loss) variations.

The revised figure showing the more number of data points is shown below (Fig. R6) with the respective logic truth table for XOR and XNOR features shown by the data.

Figure R6. (a) and (b) Experimental transmission spectra shown for increasing and the decreasing configuration in $|\Delta V|$, with increasing V_1 by keeping $V_2 = 0$ and by increasing V_2 while keeping $V_1 = 35 \text{ V}$, respectively. (c) Represents the plot of measured transmission intensity of Fano resonance for increasing and decreasing pathways of differential voltage ΔV , which shows the hysteresis type of closed loop signifying the MIO configuration in the electro-optical properties. In the inset the truth-table of XOR gate is depicted that is derived from the MIO feature in the intensity states of Fano resonance for voltage inputs V_1 and V_2 . (d) Represents the plot of measured Q -factors of Fano resonance for increasing and decreasing pathways of differential voltage ΔV , which shows two distinctive pathways in its

variation signifying the MIO configuration. In the inset the truth-table of XNOR gate is depicted that is derived from the MIO feature of the observed Q -factors of Fano resonance for voltage inputs V_1 and V_2 .

2. The reviewer felt the explanations for NOT operation (Fig. 4) can be briefed since it is included in XOR table with fixing either $V_1 = "1"$ or $V_2 = "1"$.

We thank the reviewer for this comment. In general, the system exhibiting the XOR functionality can also be used as the negation (NOT) operator by closing the input possessing high ('1') logic state and other input being varied. This condition leads to the inverter or the NOT gate possessing high contrast output intensity. On the other hand, by closing the input having the low ('0') logical state, the buffer/pass gate can be realized, where the operation does not alter the input state and output logic state is same as the input logic state. This comes for the advantage of possessing the XOR operation using the proposed MEMS Fano metamaterial that is nontrivial to realize using many photonic systems. In our work, the presented NOT operation signifies the formation/annihilation of a new resonance mode (Fano mode) in the absence/presence of external stimulus in the coupled metamaterial system. This negation operation in the Fano resonance feature represents the switching between the coupled and uncoupled regime of the near-field interactions in the absence and presence of external stimulus.

To clear the ambiguity, we have moved the figure representing the NOT gate from the manuscript to the Supplementary Information Fig. S14. Instead, we further extended our investigation to prove the presence of XNOR gate functionality (Fig. R6 (d)) in the form of far-field Q -factors (loss-factors), which is also referred to as 'equivalence gate' that is mainly used in TTL and CMOS ICs to avoid the high output resistance and the heating effects.

3. By referring supplementary, the reviewer could understand the Al thickness can tune the released height. However, still it was not clear what the exact reason of choosing 900 nm-thick Al for the actual fabrication.

Answer:

We thank the reviewer for an excellent question. We totally agree with the reviewer on his /her statement that the released heights of the cantilever can be tuned by varying in the thickness of the aluminium metal^[7]. The samples were fabricated with other different

thicknesses (300nm, 500nm and 700nm), but the main reason for using the 900nm thick aluminium resonator sample for the measurements was the better stability of the thick cantilevers for a *continuous/persistent tuning* of their released height to experimentally realize the gradual tuning of Fano resonance feature in the structure. In the Supplementary Information Fig. S4, we have included the measured data for all the metal thicknesses (300nm, 500nm and 700nm) of the samples that show the symmetric (both released configurations), maximum asymmetric (one released and other in snapped configuration) and symmetric state (both in snapped configurations) of the devices (Fig. R7 (a-c)). We have included a small information on the metal thickness dependent spectra in the page 5, paragraph 1 of revised manuscript. The electro-optical response from all these samples show the similar behavior in terms of resonant characteristics, except gradual redshift and different strength of Fano resonance features. For 300 nm thick metal sample, the release height will be greater than the metal thicknesses of 500 and 700 nm samples and hence it exhibits maximum asymmetry and maximum strength in the amplitude of Fano resonance, as shown in Fig. R7 (d). The red shift seen in the resonances of the samples for increasing metal thicknesses is due to increase in the capacitance for the thicker metal sample possessing

smaller released heights.

Figure R7. (a)-(c) Depicts the experimentally measured transmission spectra for different thicknesses (300, 500 and 700 nm) of aluminium (Al) SRR cantilevers (SRR-1 and SRR-2) in both released states ($\delta = 0$), one released-one snapped state ($\delta = \delta_{\max}$) and both in the snapped states ($\delta = 0$), respectively. (d) Shows the measured transmission spectra for various thicknesses of Al cantilevers (300, 500 and 700 nm) at the maximum asymmetry state ($\delta = \delta_{\max}$) of the sample. 300 nm thick metal sample exhibits stronger Fano resonance amplitude as it exhibits maximum released height and hence maximum asymmetry.

4. Is the multiple input more than 3 or 4 possible? If we can increase the number of the input in the future, that would be very interesting and more functional.

Answer:

This is a brilliant suggestion as a future direction for the authors to consider. We agree with the reviewer's views and perspective of using multiple inputs (> 2) in the unit cell that could lead to intriguing and more versatile functionalities of the metamaterial devices^[8]. For example, the super-cell consisting of four SRR resonators can be operated using four

independent voltage controls, which is possible to realize by performing two layer fabrication

process. This would enhance the functionality of the design by enabling the possibility of realizing NAND, OR logical operations and other output sequences by performing the permutation of all four input controls. Hence, our design showing many intriguing properties such as multiple-input-output states resembling the hysteresis-type of closed loop in the electro-optical properties and versatility of the device in displaying nontrivial logical gates at sub-terahertz bandwidths could greatly benefit future generation of digital and programmable metamaterials.

5. Some characters are too small in Figs. 3(c) and (d).

Answer:

We thank the reviewer for suggestions on the figures. We have revised the figures with much better clarity and visibility of the given content. Fig. 3(c) and 3(d) are given as Fig. 4 (a) and 4(b) in the revised manuscript.

References

1. W. Y. Kim et al., *Nature Communications*, **7**, 10429, 2016.
2. O. Buchnev, N. Podoliak, T. Frank, M. Kaczmarek, L. Jiang, and V. A. Fedotov, *ACS Nano*, **10**, 11519–11524 (2016).
3. A.J.L. Adam, *J Infrared Milli Terahz Waves*, **32**: 976 (2011).
4. O. Mitrofanov, L. Viti, E. Dardanis, M. C. Giordano, D. Ercolani, A. Politano, L. Sorba, and M. S. Vitiello. *Sci. Rep.* **7**, 44240; (2017).
5. Y. Sang, X. Wu, S. S. Raja, C.Y. Wang, H. Li, Y. Ding, D. Liu, J. Zhou, H. Ahn, S. Gwo, J. and Shi, *Adv. Opt. Mater.*, 1701368, (2018).
6. Y. Bai, K. Chen, T. Bu, and S. Zhuang, *Journal of Applied Physics*, **119**, 124505 (2016).
7. P. Pitchappa, C. P. Ho, L. Dhakar, Y. Qian, N. Singh, and C. Lee, *JMEMS Letters*, **24** (3), 525 – 527 (2015).

8. Zheludev, N. I. Obtaining optical properties on demand. *Science* **348**, 973–974 (2015).

Reviewers' comments:

Reviewer #1 (Remarks to the Author):

I have reviewed the revised manuscript and read authors' response to comments and suggestions. Authors have spent significant amount of effort to address all reviewer's comments which made the manuscript even stronger. This manuscript can now be published in Nature Communications without any further revision.

Reviewer #2 (Remarks to the Author):

After reading the bunch of responses and the pretty steady extra investigations in the supplementary, now I have convinced with the unique property that is different from the previous works and maybe its future possibility in terms of the functionality more than the logic operations by combining with other physics. Then in order to encourage the micro-opto-electro-mechanical field, I would recommend this manuscript would be published in Nature Communication with optional modifications.

Let me ask some extra questions which may lead to minor modifications;

1) The attached video demonstrating the device operation was intuitive and attractive. However, this reminds me of the switching time issue (in the video, all the cantilevers bends quite simultaneously, but recovered so randomly, then the transmission change seems to be slow). I could not find any excuses for that in the current manuscript. Even in the application like Fig. S18, such slow modulation must be inconvenient. By miniaturizing or modifying the MEMS/NEMS design and changing the operation frequency to the higher side, how fast modulation speed do the authors expect finally? Please address to this.

2) Thank you for suggesting how to cascade the logic gates as illustrated in Fig. 5R. So basically in between each gate, O-E-O conversions are necessary. If this is true, it seems we cannot avoid the system complexity that degrades the versatility. In order to recover the signal contrast, I believe we need O-E-O repeaters at some specific points in order to realize unique functionalities. But it becomes unreasonably costly if we need to put them a lot.

3) It may be too late to ask, but in terms of the criterion of far-field type logic operations, do we always need to refer the spectral response to decide whether $F = 0$ or 1 ? To simplify this decision processes, maybe we can use the bandpass filter and measure the transmittance around Fano resonance frequency. However, in this case, the contrast between " $F = 0$ " and " $F = 1$ " seems to be not good from the transmittance changes. Are there any ideas to enhance the signal contrast? Anyways to digitize the operation results, we definitely need some thresholding process though. This issue is also relevant to the cascading issue as discussed above.

Response Letter to the Reviewers

We thank the reviewers for their time and precise reading of the manuscript and for their valuable comments and suggestions. All the comments from the reviewers have been answered with greater attention and details and the same is highlighted in blue font. The revisions in the manuscript are highlighted with yellow background.

Reviewers' comments:

Reviewer #1 (Remarks to the Author):

I have reviewed the revised manuscript and read authors' response to comments and suggestions. Authors have spent significant amount of effort to address all reviewer's comments which made the manuscript even stronger. This manuscript can now be published in Nature Communications without any further revision.

We thank the reviewer for his/her appreciation our work and for recommending the manuscript for the publication at *Nature Communications* without further modifications. We further acknowledge reviewer's precise reading and for his/her insightful and constructive comments on the manuscript.

Reviewer #2 (Remarks to the Author):

Response to Authors

After reading the bunch of responses and the pretty steady extra investigations in the supplementary, now I have convinced with the unique property that is different from the previous works and maybe its future possibility in terms of the functionality more than the logic operations by combining with other physics. Then in order to encourage the micro-opto-electro-mechanical field, I would recommend this manuscript would be published in Nature Communication with optional modifications.

We express our sincere gratitude to the reviewer for positively analyzing our responses and for supporting our unique and novel results showing the multifunctional features and logic functionalities of the proposed structure. We also thank him/her for recommending the manuscript for publication in *Nature Communications* with optional corrections.

Let me ask some extra questions which may lead to minor modifications;

1) The attached video demonstrating the device operation was intuitive and attractive. However, this reminds me of the switching time issue (in the video, all the cantilevers bends quite simultaneously, but recovered so randomly, then the transmission change seems to be slow). I could not find any excuses for that in the current manuscript. Even in the application like Fig. S18, such slow modulation must be inconvenient. By miniaturizing or modifying the MEMS/NEMS design and changing the operation frequency to the higher side, how fast modulation speed do the authors expect finally? Please address to this.

We thank the reviewer for raising the point on the speed of the MEMS/NEMS design. In the MEMS actuation video showing the repeatable operation (included with our earlier response letter), the operation of the device was carried out at lower switching frequencies in the range of few tens of hertz for the purpose of clarity in capturing the video. The maximum speed of the cantilever, in principle is dictated by the resonance frequency that governs the device speed. As an example, the devices consisting of 200 nm/1 nm thick Al/Cr bimorph cantilevers of length 10 μm (similar to dimensions of our design) are shown to be operated at much faster speeds of nano-seconds (120 ns)^[1] that corresponds to 8.5 MHz of mechanical resonance frequency of the MEMS cantilever. The mechanical resonance frequency of the cantilevers that is dictated by the size, length and thickness of the metal cantilevers is calculated using the following expression^[2],

$$f_n = \frac{1}{2\pi} \left(\frac{C_n}{L} \right)^2 \sqrt{\frac{(EI)_{\text{eff}}}{\rho A}} \quad (1)$$

Here, $(C_n/L) = k_n$ is the wave number of the n th mode of the cantilever, L is the length of the bimorph cantilever, $(EI)_{\text{eff}}$ is the flexural rigidity that depends on the Young's Modulus (E) and ρA is the linear mass density of the bimorph cantilever. Considering the length of our designed cantilever that is $L = 25 \mu\text{m}$ and by using the values given in Ref [1], the theoretical resonance frequency of our bimorph cantilever (Al/Al₂O₃) is calculated to be around 3 MHz that corresponds to the maximum achievable switching speed of 330 ns. This speed can be further increased by reducing the length of the MEMS cantilever.

In the newly attached Supplementary Video-2 (volatile/repeatable operation of the cantilevers), the repeatable operation of the fabricated MEMS Fano-metasurface device is shown, where the device was operated at 100 Hz.

2) Thank you for suggesting how to cascade the logic gates as illustrated in Fig. 5R. So basically in between each gate, O-E-O conversions are necessary. If this is true, it seems we cannot avoid the system complexity that degrades the versatility. In order to recover the signal contrast, I believe we need O-E-O repeaters at some specific points in order to realize unique functionalities. But it becomes unreasonably costly if we need to put them a lot.

We agree with the reviewer on this comment, where he/she points out that the O-E-O conversions are necessary for the cascading operation, which is solely due to the fact that the device is driven electrically and the response is detected in the THz spectral domain. However, in the present device configuration, using multiple input channels (greater than two inputs) would provide more functionality of the device and hence would avoid the cascading of the devices that ought to realize other logical operations. Moreover, the O-E-O conversion will not be affecting the efficiency of the cascading operation (it can be made more efficient), as each element in the sequence will act as a new triggering source for the subsequent elements (device) driven by the constant voltage source. Unlike the devices purely operating in the electrical or the optical domains, the proposed MEMS Fano-metasurface device with O-E-O configuration is scalable and provides a reliable platform for realizing multifunctional on-chip based electro-optical devices across the wide spectrum of electromagnetic domain ranging from low THz to the optical frequencies.

3) It may be too late to ask, but in terms of the criterion of far-field type logic operations, do we always need to refer the spectral response to decide whether $F = 0$ or 1 ? To simplify this decision processes, maybe we can use the bandpass filter and measure the transmittance around Fano resonance frequency. However, in this case, the contrast between “ $F = 0$ ” and “ $F = 1$ ” seems to be not good from the transmittance changes. Are there any ideas to enhance the signal contrast? Anyways to digitize the operation results, we definitely need some thresholding process though. This issue is also relevant to the cascading issue as discussed above.

We thank the reviewer for this excellent question. We would like to clarify that for realizing the far-field logic operation it does not always require to measure the entire far-field spectral response of the device. This also can be achieved by using the band-pass filter in the far-field and measure the transmittance response, as suggested by the reviewer. Additionally, instead of the terahertz time domain (THz-TDS) source (used in our measurements, that measures the

THz-time pulse), a single frequency source (continuous wave THz source) can be employed as the read-out THz beam and the out-put power can be measured from the photo-detector, which can show a good contrast in terms of change in transmittance. From our measured transmittance data using the THz-TDS readout, the change in the transmittance is measured to be around $\Delta T = 0.61$ ($T_{\text{OFF}} = 0.71$ for $F = 0$ state and $T_{\text{ON}} = 0.1$) between the ON ($F = 1$) and OFF ($F = 0$) states, as shown in the following figure R1. This output contrast is significantly higher than the other logic gates shown in the THz ($\Delta T \approx 0.2$)^[3] and in optical frequencies ($\Delta T \approx 0.25$)^[4]. We have included the plot showing the intensity contrast in the logic outputs at frequency 0.56 THz in the Supplementary Information Figure S7. The observed intensity contrast in the ON/OFF Fano states of the device is discussed briefly on page 8, paragraph 1 of the revised manuscript.

Therefore, with higher contrast ratios in the optical output logic signals, an efficient cascading is possible by defining optimal threshold levels in the input and the output signals. Thus, making it an efficient metamaterial design for realizing high contrast logic operations in the terahertz frequencies and by scaling the device this concept can be further extended to higher frequencies without compromising on the output signal contrast.

Figure R1: Measured THz transmittance spectrum of the fabricated MEMS Fano-metasurface of metal thickness 900 nm showing the high output intensity (transmittance) contrast (ΔT) between the OFF ($F = 0$) and the ON ($F = 1$) logic states of the device. The measured output signal contrast is about $\Delta T = 0.61$ at 0.56 THz frequency.

References:

1. S. W. Lee, S. J. Park, E. E. B. Campbell & Y. W. Park. A fast and low-power microelectromechanical system-based non-volatile memory device. *Nat. Commun.* **2**, 220 (2011).
2. S M C Abdulla, H Yagubizade and G J M Krijnen, Analysis of resonance frequency and pull-in voltages of curled micro-bimorph cantilevers. *J. Micromech. Microeng.* **22**, 035014 (2012).
3. W. Y. Kim, H.-D. Kim, T.-T. Kim, H.-S. Park, K. Lee, H. J. Choi, S. H. Lee, J. Son, N. Park and B. Min, Graphene–ferroelectric metadevices for nonvolatile memory and reconfigurable logic-gate operations', *Nature Communications* **7**, 10429 (2016).
4. M. Papaioannou, E. Plum, J. Valente, E. T. F. Rogers and N. I. Zheludev. All-optical multichannel logic based on coherent perfect absorption in a plasmonic metamaterial. *APL Photonics* **1**, 090801 (2016).

REVIEWERS' COMMENTS:

Reviewer #2 (Remarks to the Author):

I could read that the authors made further efforts to address my additional questions. The manuscript is well-polished and now seems to be great enough to be published as it is (or with some minor modifications for formatting the figures).

Response Letter to the Reviewers

We thank all the reviewers for their time and precise reading of the manuscript and for their valuable comments and suggestions. In the following, all the comments from the reviewer have been answered with greater attention and details and the same is highlighted in blue font.

REVIEWERS' COMMENTS:

Reviewer #2 (Remarks to the Author):

I could read that the authors made further efforts to address my additional questions. The manuscript is well-polished and now seems to be great enough to be published as it is (or with some minor modifications for formatting the figures).

We thank the reviewer for his/her suggestions and comments on the manuscript and recommending it for publication at *Nature Communications*. We have considered all his/her comments and revised the manuscript with necessary formatting of the figures to comply with the templet of *Nature Communications*.